# Functional Ultrastructure of the Excretory Gland Cell in Zoonotic Anisakids (Anisakidae, Nematoda)

**DOI:** 10.3390/cells8111451

**Published:** 2019-11-17

**Authors:** Ivona Mladineo, Jerko Hrabar, Hrvoje Smodlaka, Lauren Palmer, Kristen Sakamaki, Kleoniki Keklikoglou, Pantelis Katharios

**Affiliations:** 1Institute of Oceanography and Fisheries, 21000 Split, Croatia; hrabar@izor.hr; 2Western University of Health Sciences, College of Veterinary Medicine, Pomona, CA 91766, USA; hsmodlaka@westernu.edu; 3Marine Mammal Care Center Los Angeles, San Pedro, CA 90731, USA; lpalmer@marinemammalcare.org; 4Pacific Marine Mammal Center, Laguna Beach, CA 92651, USA; ksakamaki@pacificmmc.org; 5Hellenic Centre for Marine Research, Heraklion, 71500 Crete, Greece; keklikoglou@hcmr.gr (K.K.); katharios@hcmr.gr (P.K.)

**Keywords:** *Anisakis pegreffii*, *Pseudoterranova azarasi*, excretory cell, microscopy

## Abstract

Excretory and secretory products are crucial for parasite infectivity and host immunomodulation, but the functioning and ultrastructure of the excretory gland cell (EC) that produces these products are still scarcely understood and described. In light of growing reports on anisakiasis cases in Europe, we aimed to characterise the EC of larval *Anisakis pegreffii* and adult *Pseudoterranova azarasi*. In the latter, EC starts 0.85 mm from the head tip, measuring 1.936 × 0.564 mm. Larval EC shows a long nucleus with thorn-like extravaginations toward the cytoplasm, numerous electron-dense and -lucent secretory granules spanning from the perinuclear to subplasmalemmal space, an elevated number of free ribosomes, small, spherical mitochondria with few cristae and a laminated matrix, small and few Golgi apparatuses, and few endoplasmic reticula, with wide cisternae complexes. Ultrastructure suggests that anaerobic glycolysis is the main metabolic pathway, obtained through nutrient endocytosis across the pseudocoelomic surface of the EC plasmalemma and its endocytic canaliculi. Thorn-like extravaginations of EC karyotheca likely mediate specific processes (Ca^2+^ signaling, gene expression, transport, nuclear lipid metabolism) into the extremely wide EC cytosol, enabling focal delivery of a signal to specific sites in a short time. These functional annotations of parasitic EC should help to clarify anisakiasis pathogenesis.

## 1. Introduction

The excretory system of the model nematode *Caenorhabditis elegans* orchestrates osmotic and ionic regulation and metabolic waste elimination analogous to the renal system of higher animals [1]. Four specialized cell types form the system: a pore cell, a duct cell, a canal cell (or excretory cell), and a fused pair of gland cells. All except the excretory gland cell (EC) are involved in the collection and excretion of fluids. The system of canaliculi most likely concentrates and transports materials from the hypodermis and extracellular pseudocoelom to the central canal for excretion [2]. The exact function of EC in *C. elegans* is unknown, but since its ablation does not lead to any defects, it seems that it is not crucial for nematode survival, e.g., molting, in contrast to the EC role in other counterparts of the phylum Nematoda [3].

Another type of excretory system consists of a single ventral gland cell or renette situated in the body cavity or pseudocoelom, which connects directly to the ventral secretory-excretory pore [4]. The duct connecting to the pore is lined with cuticle only at its distal end. In some species, the duct is looped within the tissue of the ventral gland, and at the point of connection there may be a dilation or ampulla. This type is found in Adenophorea, i.e., nematodes without phasmids, which include many free-living, marine and fresh-water species. However, there is a general consensus that the differences in the excretory system found in nematodes, both at the taxonomic and developmental levels (e.g., larval vs adult forms) might be introduced by differences in its functions [3], as well as the physiological state of the host environment [5].

In parasitic Nematoda, EC is tightly related to the pathogenicity of the species, as its function encompasses production, excretion, and secretion of a multitude of bioactive excretory and secretory products (ES) [6]. Some of these products that have hydrolytic activity engage in enzymatic degradation of host tissues, enabling larval penetration and migration, and potentially feeding [7,8]. Other important functions of ES include antigenic interaction with the host immune system that elicits an allergic reaction [9,10], modulation of T helper and innate immunity axes [11,12], larval ecdysis [13], and as recently suggested, antimicrobial activity within the host gastrointestinal system [14,15,16,17]. The exact nature and type of action have been identified only for a handful of ES products recently boosted by application of proteomic and transcriptomic tools [18,19,20,21].

The zoonotic nematode *Anisakis* spp. that parasitizes cetaceans as a final host, fish and cephalopods as paratenic hosts, and accidentally humans during its third-larval stage (L3) has an EC situated in the esophageal region, being recognized as the major site responsible for larval infectivity [22]. In in vitro cell line models, *Anisakis pegreffii* ES products have been observed to activate kinases important for inflammation, cell proliferation (or inhibition of apoptosis), and induction of p53, negatively affecting the host DNA [23,24]. In contrast, *Anisakis* spp. crude extract (CE) rather than ES products, is capable of enhancing in vitro cell proliferation, suppressing apoptosis, and inducing changes in the expression of serum cancer-related miRNAs in rats, suggesting its tumorigenic potential [25]. While new tools and methodologies have enabled insights into properties and function of *Anisakis* spp. ES products, basic knowledge on EC ultrastructure is limited [26,27].

Using high-pressure freezing fixation technique and transmission electron microscopy (TEM), in conjunction with 2D and 3D micro-computational tomography (µ-CT), our aim was to describe the ultrastructure of anisakid EC that would help elucidate its functional traits. Results obtained through advanced microscopy techniques provide a basis for functional annotation of EC properties in the future.

## 2. Materials and Methods

### 2.1. Sampling and Identification of the Larval and Adult Anisakidae

Vigorously moving third-stage larvae (L3) of *Anisakis* spp. were collected from the visceral organs of blue whiting (*Micromesistius poutassou*) caught off the Central East Adriatic Sea coast (Food and Agriculture Organization European fishing zone 37.2.1). For each microscopy technique, a subsample of five larvae was processed according to the protocol explained below. Larvae were collected, washed in filtered and autoclaved seawater (SW), and a subsample was identified morphologically as *Anisakis* sp. under light microscopy (SZX10, Olympus, Tokyo, Japan) based on known species geo-distribution in the Adriatic Sea [28]. Molecular confirmation was done following Mladineo et al. [28]. Briefly, genomic DNA was isolated in a subsample of ten larvae, and the mitochondrial cytochrome oxidase 2 (cox2; ∼600 bp) locus was amplified using PCR parameters described previously [29].

Because of scarce availability of freshly stranded carcasses of toothed whales in the Adriatic Sea, no adult *Anisakis* spp. specimens adequate for microscopy techniques were available; therefore, adult Anisakidae were collected from Californian sea lions (*Zalophus californianus*). The specimens were collected during routine necropsy of newly deceased stranded California sea lions that died during rehabilitation at the Marine Mammal Care Center Los Angeles, San Pedro, California, USA. The specimens originated from the population of California sea lions off the Pacific coast of southern California. The specimens were fixed in 70% ethanol and/or 2% paraformaldehyde and were shipped to Institute of Oceanography and Fisheries, Split, Croatia, for further assessment. Molecular identification was done by amplification of the cox2 locus as described above.

### 2.2. Confocal Microscopy of the Excretory Cell of Anisakis sp. Third-Stage Larvae

To position *in situ* the excretory cell (EC) within the body cavity of *Anisakis* sp. L3, cytoskeletal proteins ß-tubulin and actin were labeled by monoclonal antitubulin-Cy3 antibody (Sigma-Aldrich, St. Luis, MO, USA) and Atto 488 Phalloidin dye (Sigma-Aldrich), respectively, while DNA labelling was done using 4,6-diamidino-2-phenylindole (DAPI) (Invitrogen, Carlsbad, CA, USA). The larval fixation protocol and microscopy (TCS SP8 X confocal microscope, Leica, Wetzlar, Germany) followed Mladineo et al. [30]. 

### 2.3. Ultrastructure of the Excretory cell of Anisakis sp. Third-Stage Larvae

Anisakid L3 larvae were cut with 1 mm biopsy punches (Integra Miltex, Plainsboro, NJ, USA) into three parts: anterior end with pharynx, ventriculus, and anterior part of intestine. Each section was placed on a metal disc and high pressure-frozen in the presence of 20% BSA in EM PACT2 (Leica Microsystems, Vienna, Austria). Afterwards, the sections were subjected to freeze substitution in 2% OsO_4_ in acetone at −90 °C for 96 h. The temperature was then raised to −20 °C (5 °C/h) for 24 h. Finally, the temperature was raised to 4 °C (3 °C/h) and maintained for an additional 20 h.

The samples were washed in acetone (3 × 15 min), infiltrated for 1 h in 25, 50, and 75% mixtures of low-viscosity Spurr resin (SPI Chem, West Chester, PA, USA) and anhydrous acetone, and then left in pure resin overnight, transferred to embedding molds, and polymerized for 48 h at 60 °C.

Semi-thin sections cut at 0.5 μm thickness were stained with 1% toluidine blue and observed under a light microscope for orientation. Ultrathin sections cut at 0.07 μm thickness were placed on formvar coated single slot grids, contrasted in ethanolic uranyl acetate (30 min) and lead citrate (20 min), and observed under a JEOL 1010 TEM (JEOL, Akishima, Tokyo, Japan) operating at an accelerating voltage of 80 kV. Images captured with a Mega View III camera (Olympus Soft Imaging Solutions GmbH, Münster, Germany) were assembled and annotated in PhotoShop CS5 software (Adobe Systems, San Jose, CA, USA).

### 2.4. Micro-Computational Tomography (µ-CT) of the Excretory Cell of Larval and Adult Anisakidae

Five adult specimens of anisakids sampled from Californian sea lions were fixed in cold TEM-grade-buffered 2% paraformaldehyde in PBS and dehydrated to 70% ethanol for 3 days before scanning at the Hellenic Centre for Marine Research (HCMR). The contrast between the soft tissues was enhanced using 0.3% phosphotungstic acid (PTA) in 70% ethanol [31]. The µ-CT (SkyScan 1172 micro-CT scanner; SkyScan, Bruker, Belgium) uses a tungsten X-ray source with an anode voltage ranging from 20 to 100 kV, 11 PM CCD camera (4000 × 2672 pixel) and a maximal resolution of <0.8 μm/pixel. Specimens were scanned at a voltage of 59 kV and 167 μA with an aluminum filter for a full rotation of 360° at the highest camera resolution. The projection images acquired during the scanning procedure were reconstructed into cross-section images using the SkyScan’s NRecon software (NRecon, Skyscan, Bruker, Belgium) with a modified Feldkamp’s back-projection algorithm. 3D volume renderings of the scanned specimens were created using CTVox software (CTVox, Skyscan, Bruker, Belgium) to visualize the anatomy of the EC and were assembled in figure plates using Inkscape.

## 3. Results

### 3.1. Molecular Identification of Larval and Adult Anisakids

Inferred by the mtDNA cox2 locus, ten larval specimens were identified as *A. pegreffii* (GenBank accession numbers: MN624202-MN624211) and six adult specimens as *Pseudoterranova azarasi* (GenBank accession numbers: MN624175, MN624177, MN6242191, MN624194, MN624199, MN624200). However, in order to work on entire specimens, larvae specifically fixed for microscopy were not molecularly identified. While this implies that there is a probability that these belong to other anisakid spp. present in the Adriatic [28], it is negligible, and therefore *Anisakis pegreffii* was designated as the target larval species for microscopy.

### 3.2. Confocal Microscopy of the Excretory Cell of Anisakis pegreffii Third-Stage Larvae

A single excretory cell (EC) consisted of a single DAPI-labelled nucleus measuring approximately 1.4 × 0.075 mm (Figure 1a). The cell is situated in the pseudocoelom in the lower to medium part of the upper third of the larval body, sublaterally bordering with the somatic muscular layer, and submedially with the oesophagus or ventricle wall. Confocal microscopy revealed conspicuous mesh-like labelling of EC nucleus chromatin, concentrated in planes perpendicular to the nucleus length (Figure 1b,d). Terminal endings of the EC nucleus ended in pyriform-like thickenings. β-tubulin labelled only the larval cuticle (Figure 1c). Actin labelled 2 µm-thick EC membrane or filamentous surface coat (according to Lee et al. [26]) and outlined the lumen of EC canaliculi (Figure 1e), showing a ramifying appearance of the EC cytoplasm.

### 3.3. Ultrastructure of the Excretory Cell of Anisakis pegreffii Third-Stage Larvae

The plasma membrane in contact with the pseudocoelom is enveloped by an approximately 1.25 µm thick electron-lucent proteinaceous layer (Figure 2a). At the contact surface between the plasmalemma and the proteinaceous layer, the former expresses bay- or funnel-like invaginations, and endocytosis is observed in this area (Figure 2b and insert). The area is enriched in small coated vesicles and secretory vesicles that are far less electron-dense then the secretory granules trafficking mucins.

Two main types of secretory granules (SG), similar in shape and ranging from 500 nm–1.1 μm in diameter, are present beneath the plasmalemma, but not in its intimate contact, and have markedly electron-lucent SG and more numerous, electron-dense SG. Among SG, cytosol is interspersed with abundant spherical small to medium size mitochondria (approximately 550 nm in diameter), possessing only few, mostly short and prominent cristae. The mitochondria matrix shows a conspicuous lamellar appearance (Figure 2c).

Some electron-dense SG are observed secreting contents within the EC cytoplasm, while some are less electron-dense, representing immature granules (condensing vacuoles) from the trans-Golgi network. Larger and irregularly shaped electron-lucent SG, likely containing accumulated misfolded proteins (Figure 2a), are observed budding from the electron-dense SG (Figure 2d). Small Golgi apparatus and few multivesicular bodies (MVB) or lysosomes are interspersed close to the plasmalemma at the cell periphery or between SG (Figure 2a). Ribosomes are present either as free organelles between SG or attached to small spherical cisternae of the endoplasmic reticulum (Figure 2b). Sparse fibrils are present between SG.

The nucleus shows double-membrane, thorn-like extravaginations towards the cytoplasm, with no conspicuous nuclear pores (Figure 2e). Condensed chromatin (heterochromatin) and the decondensed euchromatin are hard to differentiate, being dispersed throughout the organelle. Interchromatin granule clusters (IGC) and perichromatin granules are abundant and randomly interspersed.

Usually two nucleoli are visible centrally, in the shape of rounded ribbon-like structures, not connected to heterochromatin. Except for the nucleoli granular component of the nucleoli, no other structures (e.g., dense fibrillar component or the fibrillar center) are discernible (Figure 2f).

The excretory duct is longitudinally embedded within the EC, approximately 1 µm wide, lined by proteinaceous electron-lucent layer, approximately 500 nm thick (Figure 3a). Its plasmalemma is crisscrossed by tubular structures of varying width, observed in longitudinal and transversal planes (Figure 3b).

There are two types of canaliculi: compact and fissure-like ones that are lined with conspicuously electron-dense apical extravaginations compared to the rest of the matrix (Figure 3c and insert). Numerous processes of likely endocytosis are observed close to the lumen, and many small-sized coated vesicles are present in the matrix. Few small mitochondria and scattered small early endosome-containing interior vesicles are present (Figure 3d).

The second type of canaliculi is what Lee et al. [26] referred to as drainage tubules; they have an irregular, more rounded shape with wide bay-like invaginations, lined by thin, fuzzy glycocalyx (Figure 3e). The EC cytoplasm close to these tubules is granulated, more electron-dense, and large SG are observed expelling their contents in its vicinity.

No endo- or exocytotic vesicles are observed close to the lumen. In contrast, large buddings are observed being released into to lumen, containing granulo-fibrillar material which is also observed expelled in the lumen (Figure 3f).

### 3.4. Micro-Computational Tomography (µ-CT) of the Excretory Cell of Adult Pseudoterranova azarasi

In the selected specimen of *Pseudoterranova azarasi*, EC started 0.85 mm from the tip of the head and extended aborally until 2.786 mm of the front body, measuring 1.936 mm in length and 0.564 mm at its highest. The cell followed the convexity of the parasite body wall, being situated between the esophagus and hypodermis and showing swelled, parenchymatous-like consistency (Figure 4 and Figure 5, Appendix A). At 0.087 mm from the head tip, the excretory canal formed as a thin tube starting medially in the EC and diverting posterioventrally.

## 4. Discussion

Involvement of the excretory cell (EC) in host-parasite interactions [11,12] and infectivity [7,8] of parasitic nematodes suggests that the cell has a crucial role in species propagation and survival. It is therefore surprising to observe a lack of knowledge about the EC cell-cycle or other innate functions that maintain it. Faced with the inability to draw conclusions about the EC cell cycle from the *Caenorhabditis elegans*, the free-living model that shows a more complex excretory system reminiscent to that in higher animals, basic knowledge for non-model parasitic nematodes, can be inferred from ultrastructural details of specific EC. However, useful descriptive data have only been obtained for a few parasitic nematodes [26,32,33,34,35,36].

In general, Ascaridoidea possesses a mononucleate, single excretory gland cell enveloping an intracellular excretory duct that continues in transverse canal connected by a median duct to the excretory pore. This has been observed in the infective larval stage of the intestinal dog nematode *Toxocara canis*, which is considered the prototype of visceral *larva migrans* seen in humans [35] and *Anisakis simplex* [26,33,37]. Since more advanced microscopy approaches available today enable new details of EC structure and consequently suggest its potential functional traits, we employed TEM and µ-CT to assess functional ultrastructure of *Anisakis pegreffii* third-stage larvae (L3) and adult *Pseudoterranova azarasi* EC. However, since only µ-CT was used in reference to the adult anisakid, we are not able to deeply elaborate on its functioning in the final host as the technique allows only in situ visualisation of the cell; rather, it serves as a reference of a cell’s dimension and appearance, helpful for comparison of the cell in larval forms.

The most striking features of the larval EC include a long nucleus with thorn-like extravaginations toward cytoplasm, numerous electron-dense and -lucent secretory granules that span from the perinuclear to subplasmalemmal area of cytoplasm, elevated number of free ribosomes compared to those attached to the endoplasmic reticulum, small and spherical mitochondria with few cristae and laminated matrix, small and few Golgi apparatuses, and few endoplasmic reticula, but with a wide complex of cisternae. These findings are mostly in line with the only existing ultrastructural study of the species [26].

The plasmalemma is enveloped by a proteinaceous sheet, which is thicker medially where it contacts the pseudocoelom than laterally where it is overlaid by somatic muscles. Interestingly, only the plasmalemma surface in contact with the pseudocoelom shows tight but branching invaginations associated with small coated vesicles. As also observed by Lee et al. [26], this is likely EC’s primary site of nutrient absorption, as small coated vesicles are possibly the result of endocytosis and membrane internalization, similar to that in the apical cytoplasm of pancreatic cells [38]. Moreover, electron-dense and -lucent secretory granules (SG) are not in intimate contact with the plasmalemma, supporting the assumption that this area is designated only for nutrients and possibly exchange of other molecules. The role of the proteinaceous coat of the plasmalemma remains unclear; it does not support excretion of excretory/secretory (ES) products from secretory granules directly in the pseudocoelom, but most likely protects EC in situ. Abundant immunohistochemical labelling of actin suggests its role in EC shape maintenance and mechanical/ chemical protection from stimuli deriving from pseudocoelom or somatic muscles. In most animal cells, however, a dense cross-linked meshwork of hundreds of actin filaments is found bound beneath the plasmalemma, building up a cellular cortex that drives shape changes during cell migration, division, and tissue morphogenesis [39]. Lee et al. [26] suggested that this actin-labelling coat could act as a filter for solid or fluid material that is absorbed in the EC, as well as a supportive element.

A system of canaliculi seems to be additionally involved in EC nutrient absorption. Lee et al. [26] described these canaliculi as “draining tubules”, characterized by a “dense cytoskeletal material” and their involvement in the discharge of secretory granules products. Conspicuous dense cytoskeletal material in the apical part of canaliculi positively labelled for actin, suggesting their filtering and supportive role within the EC, rather than excretory. Furthermore, the cortical actin network as a barrier of exocytosis was recognized a long time ago [40], supporting our hypothesis that these canaliculi enable only endocytic absorption towards the EC cytosol.

In line with our observation, Lee et al. [26] failed to observe secretory granules in intimate contact with these draining tubules but observed numerous membrane-bound electron-lucent vesicles (120–170 µm). We argue that these vesicles belong to the same type of small, coated vesicles involved with endocytosis in the plasmalemma, the latter supported also by formation of endosomes containing multiple small vesicles. Whether these vesicles are involved solely in EC feeding or in osmoregulation as well still needs to be clarified. In larval *Toxocara canis* for example, the absence of osmoregulatory processes in EC has been suggested based on the lack of a complex lamellar system equivalent to the membranes of transporting epithelia [35]. In contrast, the feeding or absorption role is further supported by Parshad and Guraya [34], who detected heavy deposition of lipids (triglycerides, phospholipids and lipoproteins) within “the walls of lateral excretory canals” and glycogen within its lumen, in line with our thesis that these previously called “draining tubules” or “excretory canals”, are in fact part of the EC feeding, not excretion system. The authors speculated that the presence of lipids in the walls can be correlated with transportation of substances across the cellular membranes and might be forming their structural material which is, in some way, related to the physiology of excretion. That the process is endocytic in these canaliculi, rather than exocytic, is also supported by the intense activity of nonspecific alkaline phosphatase in the canaliculi wall, which enables the active transport of sugars against a concentration gradient from the lumen to the EC cytosol [34].

Although *Anisakis* sp. L3 are non-active feeding stages, a recent study demonstrated the presence of several vesicles in the L3 intestinal lumen, potentially originating from passive intake during migration through the tissues of their paratenic hosts [41]. It is likely that a part of these “ingesta” in the form of nutrients finally reaches the EC and passes by endocytosis, as the cell is already active at this larval stage and requires energy for ES production. Therefore, we suggest that the energy required for EC metabolism in L3 is partially obtained by endocytic processes of nutrients ongoing through the pseudocoelomic surface of the EC plasmalemma and its endocytic canaliculi.

In addition to this feeding role, L3 endocytic vesicles could also be involved in the formation of SG. Specifically, it has been observed that in some hematopoietic cells, a convergence of biosynthetic and endocytic membrane traffic analogous to lysosomal biogenesis is required for SG formation [42]. Such SG are accessible via the endocytic pathway, and their dense-core formation can occur in multivesicular bodies, observed also in L3 EC. This population of SG shares several properties with lysosomes, and since conventional lysosomes can fuse with the plasmalemma, it may be suggested that a part of SG evolved from a lysosomal progenitor [42]. The rest of SG is the product of biosynthetic pathway alone, that occurs at the trans-Golgi network (TGN), while further maturation occurs in the cytosol. That this pathway is more represented in L3 EC is evidenced by the presence of less electron-dense immature granules called condensing vacuoles.

The synthesis of ES products and building up of SG within the endoplasmic reticulum (ER) and Golgi apparatus, respectively, requires energy. Mitochondria are located mainly at sites where energy is needed, but can change their location according to requirements, as well as undergo temporary alterations in shape [38]. We observed a scarce number of small spherical mitochondria with few short cristae in the EC indicating that in L3, energy requirements are not extensive [43], or that the oxidative phosphorylation that occurs in the cristae membrane surface [44] is not necessarily the only source of energy. Alternatively, anaerobic glycolysis that mostly occurs in the cytosol in an oxygen-independent metabolic pathway converts glucose into pyruvate, storing the energy in form of ATP and NADH. Both somatic muscle and enterocytes of L3 have been observed rich in glycogen [41], suggesting that glycolysis might be an important metabolic pathway at the larval stage, engaged also in functioning of the EC.

Observed also by Lee et al. [26], EC mitochondria showed a granulated laminar matrix, which is indicative of the storage of Ca^2+^ and other divalent cations [38]. This is important as in many cell types, increased intracellular Ca^2+^ is the main trigger for granule exocytosis, followed by ATP-dependent reorganization of the cortical actin cytoskeleton, so that granules can be recruited to the plasmalemma [45]. We observed more free- then ER-bound ribosomes in the L3 EC, supporting elevated processes of mitochondria building-up that will prepare the cell for more energy-demanding processes to follow. Specifically, mitochondria-destined precursor proteins are synthesized on cytosolic ribosomes and are actively imported through the complex termed the translocase of the outer membrane (TOM), across which they reach their functional destinations inside the mitochondria [46].

Secretory granules (SG) in the L3 have been observed throughout the whole EC, except in intimate contact with plasmalemma or endocytic canaliculi. These organelles are regulated by a secretory pathway that permits exocytosis; a controlled release of granule content in response to physiological signals. It is a multistage process that may result in the release of the entire population of SG or only a small portion of them [42]. We have observed accumulation and degranulation of SG in the vicinity of irregularly shaped excretory canaliculi, which unlike endocytic canaliculi, are embedded in the dense, granulated matrix. The lumen of these excretory canaliculi is shaped by blobbing membrane-bound extravaginations that eventually release their contents by exocytosis. Apparently, L3 ES products are first discharged in the cytosol close to the excretory canaliculi by rupture of SG membrane, pressured towards the lumen and discharged through reorganization of the canaliculi membrane in the form of a blob. This greatly differs from the standard exocytosis observed in neuroendocrine or exocrine cells, where the SG membrane fuses with the plasmalemma and then releases its content outwards, directly in the lumen [42]. Whether observed exocytosis in *Anisakis* sp. is a feature characteristic only in larval stage needs to be further assessed.

The lumen of excretory canaliculi is observed moderately filled with granulo-fibrillar dense substance, while its membrane is covered with mucin-like glycocalyx, the latter essential for effective functioning of the ES system [47]. In contrast, in the main excretory duct lined by cuticle [48], we failed to observe exocytosis, but there were numerous crisscrossing structures depicted as rounded transversal and tubular longitudinal sections. It is likely that these tubular structures are in fact functional draining tubules that collect and direct ES discharged in cytosolic excretory canaliculi into the main excretory duct.

Although the content of L3 SG can most easily be identified by immuno-electron microscopy, its appearance indicates merely that the electron-lucent SG contain mucin-like highly glycosylated glycoproteins, while serous-like content appears as an electron dense material, tightly fitting limiting membrane [38].

The *Anisakis pegreffii* L3 EC nucleus shows conspicuous thorn-like extravaginations of the karyotheca and mesh-like labelling of chromatin observed by TEM and confocal microscopy, respectively. Although such a shape has not been literally described nor discussed by Lee et al. [26], it can be observed in the author’s micrographs. Interestingly, Mueller [33] also observed these nuclear processi, referring to them as “small outwardly directed folds in the dorsal nuclear wall, from the points of which proceed outward darkly staining fibrous streamers of nuclear material which frequently divide and presumably penetrate to all portions of the gland.” All of the above suggest that such a nucleus shape in L3 is physiological, although its consequences for EC functioning still remain elusive. Interestingly, in many normal and abnormal cells throughout the plant and animal kingdom, parts of the nuclear lamina (NL) extending deep inwards into the nucleus interior, called nucleoplasmic reticulum (NR), have been reported [49]. The functions of NR are still obscure, but the preliminary evidence suggests its role in Ca^2+^ signaling, gene expression, transport, and nuclear lipid metabolism. We can only speculate whether such appearance of the NL in L3 EC might help to mediate some of the mentioned processes into the extremely wide EC cytosol, therefore enabling focal delivery of a signal to specific sites within the cytosol in a short time.

Another atypical characteristic of larval nucleus is a lack of clear eu- and heterochromatin structuring and distribution, while the nucleus mostly contained interchromatin granule clusters (IGCs). In effect, such a “highly modified” [33] appearance of nucleus erroneously prompted some authors to consider it a bladder, rather than nucleus [50]. IGCs represent the ultrastructural equivalent of the splicing speckles or splicing factor compartments that play a crucial role in efficient coupling of transcription and pre-mRNA splicing [51]. While transcription and pre-mRNA splicing do not generally seem to take place within these nuclear regions, the assembly, modification, and/or storage of proteins involved in pre-mRNA processing are done in IGCs [52]. This suggests that the L3 nucleus intensively undergoes processing stages (5’ capping, splicing, 3’-end processing, and export) to deliver mature mRNA ready to translate proteins, in order to secure development and growth towards the next larval stage. This is also reflected in the nucleolus, whose structure is a result of transcription and processing of pre-rRNA and assembly of precursors of the small and large ribosome subunits. Since we have clearly observed only the granular components of L3 EC nucleolus that should contain growing pre-ribosomal particles at late stage of formation [53], we can deduce that ES production in L3 is still not at its highest, despite the presence of abundant SG within the cell.

The excretory cell of the *A. pegreffii* infective third-stage larvae shows many ultrastructural peculiarities that can direct future work towards functional annotation of processes taking place in the parasite. Molecular techniques coupled with comparative studies in other parasitic nematodes are meant to extend our understanding of EC that represents a main source of species pathogenicity.

## Figures and Tables

**Figure 1 cells-08-01451-f001:**
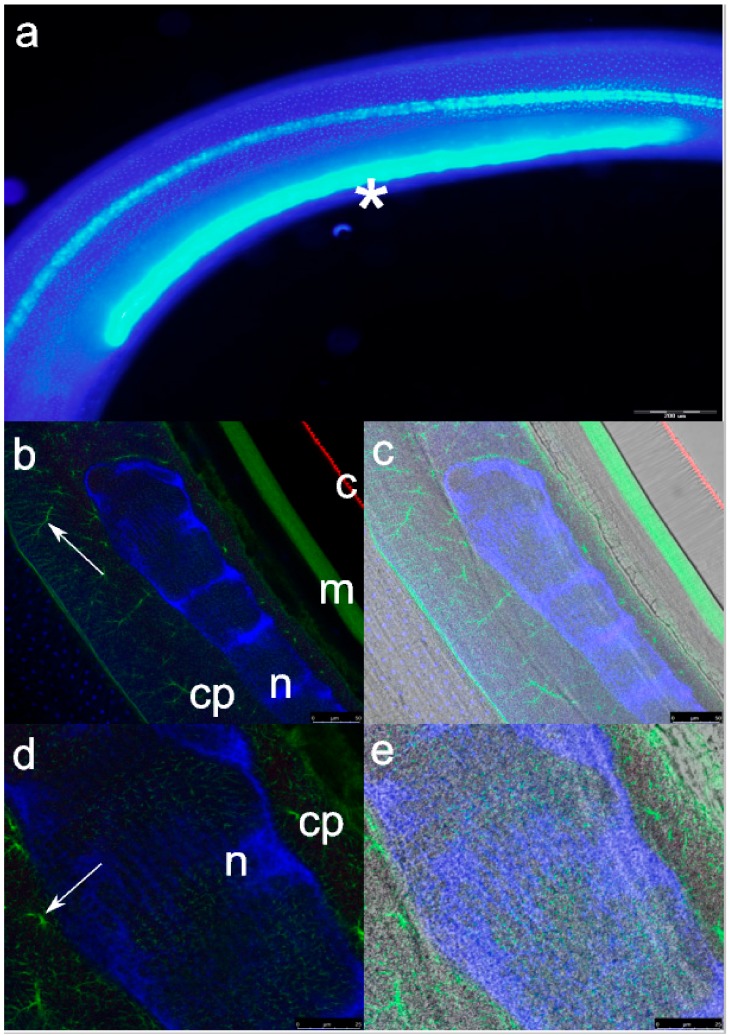
Immunohistochemical localization of β-tubulin (red, labeled by monoclonal antitubulin-Cy3 antibody), actin (green, labeled by Atto 488 phalloidin), and excretory cell nucleus (blue, labeled by 4,6-diamidino-2-phenylindole [DAPI]) in *Anisakis pegreffii* third-stage larvae: (**a**) whole-mounted specimen showing long, rod-like excretory cell nucleus (*); (**b**) excretory cell nucleus (n), cytoplasm (cp), somatic muscles (m), and cuticle (c). Arrow points to canaliculi in the excretory cell cytoplasm; (**c**) same as under (**b**) after merging of the specimen under the bright field and immunofluorescence; (**d**) higher magnification of the excretory cell nucleus (n), cytoplasm (cp), and canaliculi (arrow). Note mesh-like localization of the DAPI signal in the nucleus; (**e**) same as under (**d**) after merging of the specimen under the bright field and immunofluorescence. Scale bars: (**a**) 200 µm; (**b**) and (**c**) 50 µm; (**d**), and (**e**) 25 µm.

**Figure 2 cells-08-01451-f002:**
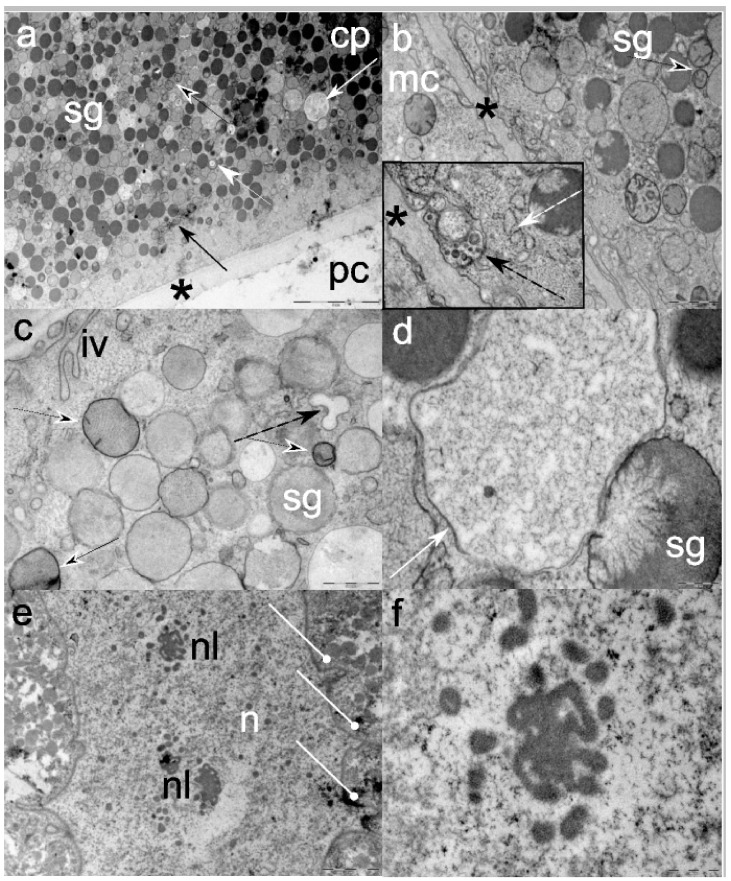
Representative electron micrographs of third-stage *Anisakis pegreffii* excretory cells: (**a**) Electron-dense serous- and electron-lucent mucin-like secretory granules (sg) in the cell cytoplasm (cp) lined by a proteinaceous layer (*) in contact with the pseudocoelom (pc). Note small Golgi apparatus (black arrow), mitochondria (black arrowhead), multivesicular body (white arrowhead), and vacuole likely filled with accumulated, misfolded products; (**b**) a cell filled with secretory granules (sg) is in close contact with a somatic muscular cell (mc) through the proteinaceous layer (*). Note small and spherical mitochondria (black arrowhead). Insert: Multiple invaginations of the plasmalemma show endocytosis and formation of endosomes with multiple vesicles (dashed black arrow). Note ribosomes on the rough endoplasmic reticulum (dashed white arrow); (**c**) plasmalemma invaginations (iv) in the cytoplasm rich with electron-lucent mucin-like secretory granules (sg), spherical mitochondria with lamellar appearance of matrix and few short cristae (black arrowhead) and empty endosomes (dashed black arrow); (**d**) electron-dense serous-like secretory granule (sg) expelling its product in a large vesicle of irregular shape (white arrow); (**e**) nucleus (n) and two nucleoli (nl). Note extravagination of nucleolar karyotheca (white round-tip arrows); (**f**) Granular component of the nucleolus. Scale bars: (**a**) 5 µm; (**b**) 1 µm (insert 500 nm); (**c**) 500 nm; (**d**) 200 nm; (**e**) 2 µm; (**f**) 500 nm.

**Figure 3 cells-08-01451-f003:**
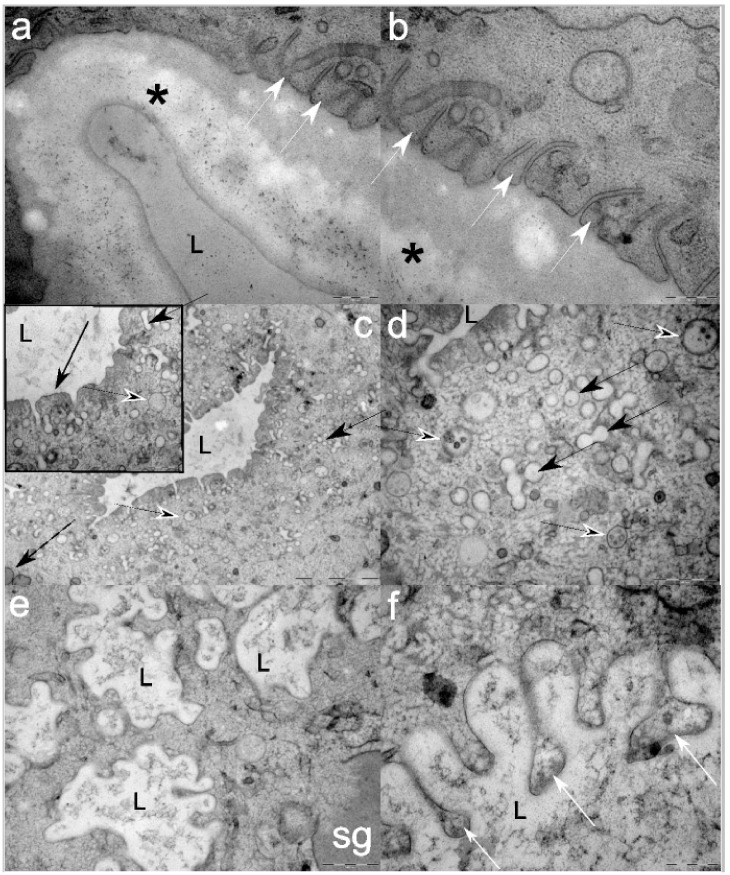
Representative electron micrographs of third-stage *Anisakis pegreffii* excretory cells: (**a**) lumen (L) of the excretory duct lined with a thick proteinaceous layer (*) and plasmalemma criss-crossed by tubular structures of varying diameters (white arrowheads); (**b**) detail of tubular (white arrowheads) structures opening beneath the excretory duct proteinaceous layer (*); (**c**) Lumen of the canaliculus (L) where intensive endocytic processes take part. Note small empty endosomes (solid black arrowhead), endosomes with interior vesicles (black arrowhead), and rare mitochondria (dashed black arrow). Insert: Detail of the lumen of the endocytic canaliculi (L) showing conspicuous electron-dense extravagination of the plasmalemma (black arrow), empty endosomes (solid black arrowhead), and endosomes with interior vesicles (black arrowhead); (**d**) Cytoplasm beneath endocytic canaliculi showing abundant empty endosomes (solid black arrowhead) and endosomes with interior vesicles (black arrowhead); (**e**) Multiple lumens (L) of excretory canaliculi (discharge tubules by Lee et al. [26]) filled with granulo-fibrillar material. Note the discharging of a large electron-dense secretory granule (sg) in the vicinity of the lumen; (**f**) Extravaginations of the excretory canaliculus showing budding of the material (white arrows) to be discharged in the lumen (L). Scale bars: (**a**) and (**b**) 200 nm; (**c**) 2 µm (insert 500 nm); (**d**) and (**e**) 500 nm; (**f**) 200 nm.

**Figure 4 cells-08-01451-f004:**
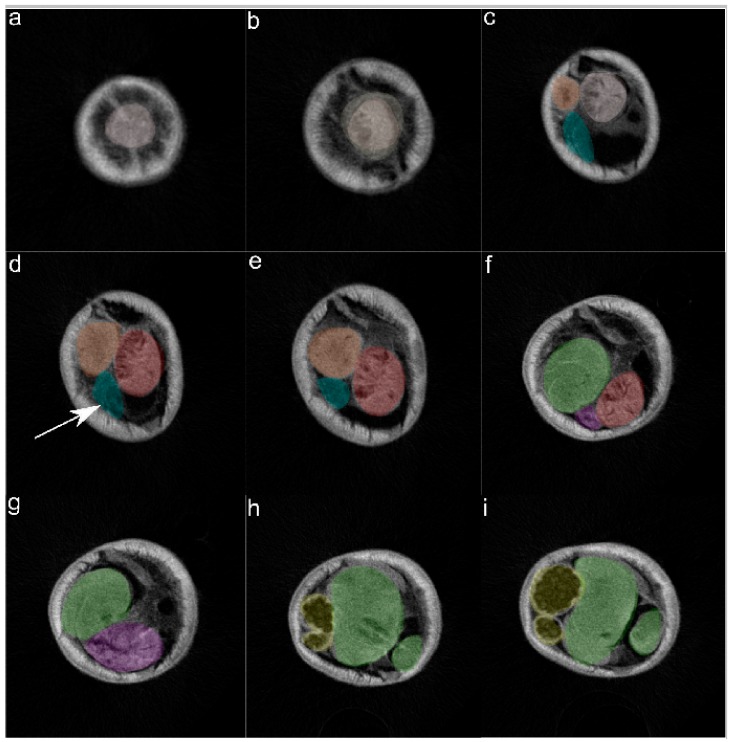
Representative transversal 2D renderings of the adult *Pseudoterranova azarasi* forebody overlaid by colors respective to different nematode organs: (**a**) esophagus (light pink); (**b**) nerve ring (light beige) surrounding the esophagus; (**c**) excretory cell (EC; blue) under-positioned by intestinal caecum (orange); (**d**) ventriculus (red) and presumptive excretory duct of the EC (arrow); (**e**) diminishing EC (blue); (**f**) ventriculus entering intestine (green) through funnel-like connection (violet); (**g**) more enhanced connective funnel (violet) flowing into the intestine (green); (**h**) and (**i**) two intestinal loops (green) and gravid uterus (yellow).

**Figure 5 cells-08-01451-f005:**
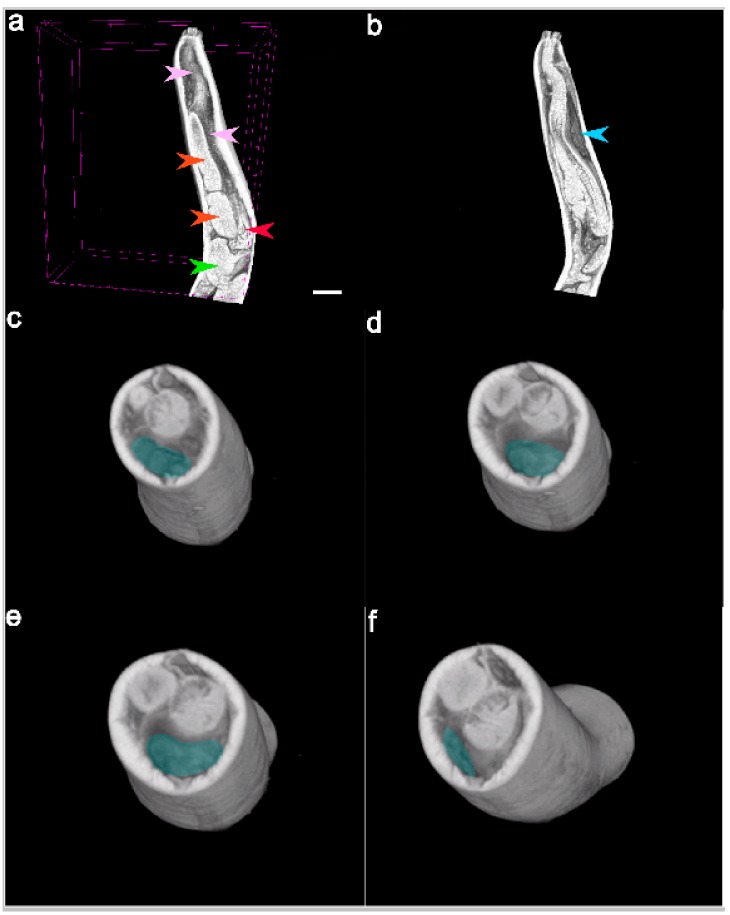
Representative longitudinal 2D (**a**,**b**) and transversal 3D (**c**–**f**) renderings of the adult *Pseudoterranova azarasi* forebody, with the excretory cell (EC) overlaid in blue color and arrows showing esophagus (pink), intestinal caecum (orange), ventriculus (red), intestine (green), and EC (blue). Scale bar: 500 µm.

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
