# Peer review of "Functional Ultrastructure of the Excretory Gland Cell in Zoonotic Anisakids (Anisakidae, Nematoda)"

_cells, 2019, doi:10.3390/cells8111451_

Round 1

Reviewer 1 Report

The manuscript provide a detailed descripton of the ultrastructure of the excretory gland cell larval adult forms of anisakid nematodes. The paper i eìs well written, scientifically sound, providing original and innovative information, useful for future studies.

Minor comments:

Delete "the" before zoonotic anisakids in the title

Use Anisakis sp. at lines 94 95 and 101 as identification at species level is a result and should not be mentioned  in the M&M section

Author Response

Dear Reviewer,

We are grateful for your overall assessment, and the suggested corrections have been inserted in the text in M&M section; A. pegreffii has been substituted by Anisakis sp. 

Sincerely,

the authors 

Reviewer 2 Report

This article addresses the microscopic description of an important zoonotic parasite and falls well within the scope of Cells. Overall, the manuscript is well written. Despite its descriptive nature, it sets the stage for future research on the involvement of excretory cell in anisakid infectivity, propagation and survival.

Nevertheless I have some concerns. Authors have conducted molecular identification of a subsample of 10 Anisakis spp larvae, being all of them A. pegreffii. Nevertheless, two additional Anisakis species, A. typica and A. ziphidarum, have been identified in Adriatic fish. Taking in consideration that the co-infection of these Anisakis species is possible, how authors can be sure that they have been working with A. pegreffii? Although it is true that A. pegreffii prevalence is higher, the possibility that they have molecularly identified A. pegreffii but they have fixed and worked with other Anisakis species cannot be ruled out. Authors should be more careful about this throughout the entire manuscript.

Authors have analyzed larval and adult Anisakidae but discussion is almost exclusively focus on larval stage. Authors should include in the discussion section more information about adult stage and the implications on the processes taking place in the final hosts.

Minor point

Line 102: how many Anisakid larvae L3 were used? Please, indicate it.

Author Response

Dear Reviewer,

We are grateful for detailed assessment and highlights of the MS's weak points. 

1. Nevertheless I have some concerns. Authors have conducted molecular identification of a subsample of 10 Anisakis spp larvae, being all of them A. pegreffii. Nevertheless, two additional Anisakis species, A. typica and A. ziphidarum, have been identified in Adriatic fish. Taking in consideration that the co-infection of these Anisakis species is possible, how authors can be sure that they have been working with A. pegreffii? Although it is true that A. pegreffii prevalence is higher, the possibility that they have molecularly identified A. pegreffii but they have fixed and worked with other Anisakis species cannot be ruled out. Authors should be more careful about this throughout the entire manuscript.

We acknowledge this possibility and have added in the text, M&M section:

However, in order to work on entire specimens, larvae specifically fixed for microscopy were not molecularly identified. While this implies that there is a probability that these belong to other anisakid spp. present in the Adriatic [28], it is negligible and therefore Anisakis pegreffii was designated as the target larval species for microscopy.

2. Authors have analyzed larval and adult Anisakidae but discussion is almost exclusively focus on larval stage. Authors should include in the discussion section more information about adult stage and the implications on the processes taking place in the final hosts.

Indeed, there is limited data on functionality of adults EC. The reason is that only µ-CT has been performed in the adults, giving information on the structure and dimension of the cell, not the ultrastructure. We have added in the text, Discussion section:

However, since only µ-CT was used in reference to the adult anisakid, we are not able to deeply elaborate on its functioning in the final host as the technique allows only in situ visualisation of the cell. Rather it serves as a reference of cell's dimension and appearance, helpful for comparison of the cell in larval forms.

3. Line 102: how many Anisakid larvae L3 were used? Please, indicate it.

We have included in the text, M&M section:

For each microscopy technique, a subsample of five larvae was processed according to the protocol further explained.